# The use of anticoagulants for rodent control in a mixed-use urban environment in Singapore: A controlled interrupted time series analysis

**Stacy Soh**[1☉], **Cliff Chua**[1☉]*, **Jane Griffiths**[1], **Penny Oh**[1], **John Chow**[2], **Qianyi Chan**[2], **Jason Tan**[2], **Joel Aik**[1,3]

1 Environmental Health Institute, National Environment Agency, Singapore, Singapore, 2 Rat Control Unit, National Environment Agency, Singapore, Singapore, 3 Pre-Hospital and Emergency Research Centre, Duke-NUS Medical School, Singapore, Singapore

☉ These authors contributed equally to this work.

* Cliff_CHUA@nea.gov.sg

**Data Availability Statement:** All relevant data are within the paper and its Supporting Information files.

## Abstract

Vector control remains an important strategy in preventing rodent-borne diseases. Studies quantifying the impact of anticoagulant bait use on rodent populations are scarce in tropical settings. This study examined the impact of anticoagulant bait use on three measures of rodent activity in Singapore to inform rodent-borne disease control strategies. Using a controlled interrupted time-series analytical design with negative binomial and linear regression models, the average rodent activity levels were compared in the pre- and post-intervention periods. There was a 62.7% (Incidence Rate Ratio (IRR): 0.373, 95% CI: [0.224, 0.620]) reduction in the number of rodents caught, a 25.8-unit (coefficient = -25.829, 95% CI: [-29.855, -21.804]) reduction in the number of 30 g/unit baits consumed and a 61.9% (IRR: 0.381, 95% CI: [0.218, 0.665]) reduction in the number of marred bait stations relative to the pre-intervention period. There was a rise in all three outcome measures within four months after the post-intervention period. This study provided strong evidence that anticoagulant baits substantially reduces rodent activity. The population resurgence after the post-intervention period reinforces the importance of timing the resumption of control measures aimed at reducing rodent-borne disease transmission.

## Introduction

Rodents are vectors for pathogens that pose threats to population health either through the direct spread to humans or indirectly via ectoparasites that thrive on them [1, 2]. One of the most important rodent-borne pathogens is *Leptospira*. Leptospirosis is caused by this bacterium and imposes a significant health burden worldwide, with the global disability-adjusted life years (DALYs) estimated at 2.9 million [3]. Approximately 1.03 million cases and 58,900 deaths occur worldwide annually, with heavier disease burdens in South and South-east Asia,

**Funding:** This study was funded by the National Environment Agency, Singapore.

**Competing interests:** The authors have declared that no competing interests exist.

Latin America and Sub-Saharan Africa. These rates are also higher in resource-poor countries where rodent control and surveillance are lacking [4].

As reservoirs for Hantavirus, aerosolised infected rodents' droppings and urine are capable of causing Haemorrhagic Fever with Renal Syndrome (HFRS) or Hantavirus Cardio Pulmonary Syndrome (HCPS), depending on the viral strain. Mortality rates of Hantavirus can go up to 12% for HFRS where it is prevalent in Asia and Europe, and 40% for HCPS, which is the prevailing strain in the Americas [5–7]. Parasitic fleas harbouring on rodents are also known to carry *Rickettsia typhi* which causes murine typhus upon human transmission. The majority of the infections occur in tropical and subtropical countries in Southeast Asia [8, 9]. Rodents, in addition to transmitting pathogens of zoonotic importance, cause significant damage and economic losses. Food contamination, broken containers and storage bags due to gnawing, as well as fires due to cable bites are some of the main consequences of the action of rodents in high densities. The use of anticoagulants is a strategy to control the population abundance of rodents, which results in a reduction in the transmission of zoonotic diseases, material damage and economic losses. For instance, Bromadiolone is a common rodenticide anticoagulant that is highly toxic and can be lethal to rodents from one day's feeding [10]. Bromadiolone disrupts the recycling of Vitamin K, preventing blood clotting, resulting in internal bleeding and eventual death [11].

Agencies rely on integrated pest management (IPM) to reduce rodent-borne disease transmission. One of the main components of IPM is focused on human behavioural change in areas of waste management and food storage. Physical, biological and chemical rodent control efforts are among other approaches aimed at reducing rodent populations [12]. Two of the most common control measures are physical trapping and chemical baiting. Physical trapping includes the use of cage traps, glue boards and snap traps while chemical baiting employs the use of attractive grain-based foods infused with anticoagulants. The consumption of anticoagulants suppresses the rodent's liver capability of synthesizing vitamin K-dependent clotting factors, which may lead to death resulting from internal haemorrhage or uncontrolled bleeding from external wounds [13, 14].

Over the years, the use of anticoagulant rodenticide as a strategy for rodent-borne disease control has become increasingly popular due to its relatively low cost and acclaimed efficacy in reducing rodent populations. However, few studies have quantified the impact of intensive anticoagulant bait use on rodent populations in tropical urban settings. In urban Singapore, licenced food establishments including restaurants, caterers, food and hawker stalls are situated all around the island, numbering in excess of 36,000. The Singapore Food Agency (SFA) is responsible for the licensing of these food establishments and all food establishments are legally mandated to obtain a license before these establishments are permitted to operate [15]. Hawker centres are food building complexes that house numerous food stalls selling a variety of inexpensive food [16]. Some hawker centres have adjoining wet markets and are located within public housing estates and business districts. These centres are accessible and affordable public communal dining spaces and are frequently patronized by locals and tourists alike [17]. Food establishments are therefore an important factor in supporting rodent population growth if food storage and food waste management practices give rise to easy food access. This study sought to quantify the impact of intensive anticoagulant use as an urban rodent population control measure, in order to inform the implementation of rodent population control strategies in Singapore.

## Materials and methods

### Ethics statement

This study was part of the National Environment Agency's existing integrated national programme of rodent surveillance and control. This study did not involve any human subjects.

The Environmental Public Health Operations Department of the National Environment Agency, Singapore (NEA) reviewed the protocols and gave study approval (EPHOD 01/2019 08012019). All applicable national guidelines for the care and use of animals were followed. Rodents trapped in the study did not belong to endangered or protected species.

## Study area

Singapore is a densely populated urban city-state located within the tropics of Southeast-Asia, with an estimated population of 5.7 million and a land area of approximately 722.5 km$^2$ [18]. Singapore is one of the most densely populated countries around the world.

This study was carried out in two urban mixed-residential and commercial use areas situated within close proximity to hawker centres and food establishments. The observations from the selected intervention site was compared with the observations with a control site that reflected variations in any secular trend which might influence the study findings.

**Intervention site (Site A).** Site A was a densely populated mixed-use development, comprising a wet market, hawker centre, and public housing apartment blocks, located within the central district of Singapore (Fig 1A). This district is a popular tourist destination and receives a high number of local visitors and tourists daily. There are numerous food establishments in this area, with one hawker centre cum market and 48 other food establishments. According to the authors' experience, there have been frequent past reports of rodent sightings and infestations in this area.

**Control site (Site B).** Site B was also a densely populated mixed-use development, comprising of a wet market, hawker centre, and public housing apartment blocks, located within in the central district of Singapore. There are numerous food establishments in this area, with one hawker centre cum market and 59 other food establishments (Fig 1B). Likewise, rodents have been a persistent issue in this district. Site B was located approximately 3.5km away from Site A and situated sufficiently far apart, with numerous natural and anthropogenic obstacles to minimize any spill over effect from the treatment site.

Waste collection frequency by licensed public waste collectors in the treatment and the control sites remained unchanged over the study period, with daily collections occurring between 7am-7pm. Likewise, the street cleaning frequency in both sites remained unchanged. There

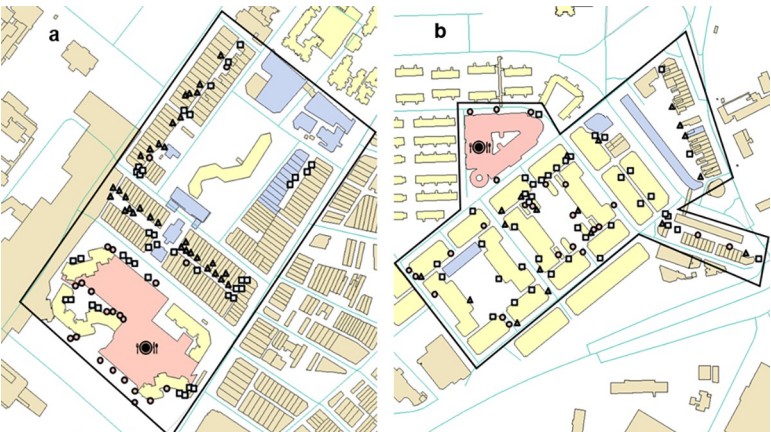

**Fig 1. Map of Site A and B respectively, with the deployment location of the cages, bait stations and hanging baits.** The square icons represent the cages, the triangle icons represent the bait stations and the circle icons represent the hanging baits. The pink shaded areas represent the hawker centres, the beige shaded areas represent shop houses, the blue shaded areas represent non-residential buildings while the yellow shaded areas represent public housing apartment blocks.

was a similar number of residential dwelling units in the intervention (550 units) and the control (517 units) sites.

## Intervention

Bell Laboratories CONTRAC BLOX, a second-generation anticoagulant rodenticide [19] containing the active ingredient Bromadiolone (0.005%), weighing approximately 30g per unit, was employed in the cages, bait stations and hanging baits in the treatment site. In the control site, Bell Laboratory's DETEX with Lumitrack baits, which do not contain the active ingredient Bromadiolone, were used.

## Outcome measures

Three outcome measures were used in the study to assess the independent effect of anticoagulant baiting on rodent activity: (i) the number of rodents trapped in cages, (ii) the amount of anticoagulant bait consumed from the cages and hanging bait stations (30g-units) and (iii) the number of marred bait stations. Bait station were defined as "marred" if there was evidence of bait consumption on at least one of the bait units within the station.

## Study design

The main study was carried out in three phases and the sites were monitored for a resurgence in rodent activity. The capture-mark-recapture (CMR) approach has been commonly used in other studies for the estimation of abundance [20, 21] and was used in this study. Removal trapping was not carried out in order to assess the independent effects of the intervention on the rodent population. Ongoing rodent control measures were maintained in both sites on an ad-hoc basis by the relevant pest control operators.

In the pre-intervention phase, the rodent population was given a 2-week period to acclimatise to the cages, baits and bait stations to overcome their food neophobia [22]. The cages and bait stations were deployed with non-toxic baits and were deactivated. After which, the cages were activated with the non-toxic baits and monitored for 4 weeks in both sites. In the intervention phase of the study, anticoagulant baits were employed only in the treatment site below ground level via hanging baits in the drain, and above ground in the bait stations, with the baits placed 5–8 metres apart (S1 and S2 Figs) for 7 weeks. Non-toxic baits were employed in the control site. Cages from the pre-intervention phase of the study were replaced with either bait stations or hanging baits depending on the viability of the location. This was done to ensure that the combined number of cages, bait stations and hanging baits in each site remained consistent throughout the whole study. Baits that showed signs of consumption were replaced daily. The post-intervention phase began with a 2-week period where non-toxic baits were deployed. After which, the cages were activated with non-toxic baits and monitored for 4 weeks. After the conclusion of the post-intervention phase, the study sites were monitored for a month to assess the extent of resurgence in the rodent population.

## Statistical analysis

A controlled interrupted time-series analytical design was used to evaluate the effectiveness of anticoagulant baiting on the outcome measures for this study. An interrupted time series analysis is a quasi-experimental study design that can be used to assess the effect of an intervention on an outcome of interest [23] and has been widely used to evaluate population level public health interventions [24]. The effect of the intervention is estimated by comparing the trend in the outcome following the intervention to the existing trend in the pre-intervention period.

The inclusion of a control group in the interrupted time series study design allows for the intervention effect to be distinguished from any secular changes in the outcome that would have occurred in the absence of the intervention [23].

Each of the three outcome measures were analysed on a daily timescale. Data collection was carried out on a daily basis on weekdays. Data was not collected on weekends, public holidays and during heavy precipitation events. Two separate negative binomial models were fitted for the number of rodents caught and the number of marred bait stations to account for overdispersion, and a multiple linear regression model for the bait consumption. The average levels and the trends of these outcome measures were estimated in the pre- and post-intervention period. Seasonality was accounted for with day, week and month variables in all three models. Additionally, the autocorrelation and partial autocorrelation function plots of the penultimate model were examined before adding lags of the deviance residuals to account for any serial autocorrelation. The general model used to examine the effect of intensive toxic baiting on the outcome measures is described in Eq (1) as follows:

$$
\begin{aligned}
log(&Outcome\ measure\ in\ Site\ A) \\
&= \beta_0 + \beta_1\ Time + \beta_2\ Intervention + \beta_3\ Time\ After\ Intervention \\
&+ \beta_4\ Outcome\ measure\ in\ Site\ B + \sum_{i,t=1,2,3,4,5}^{t=5} \beta_{5,i} Day_t \\
&+ \beta_6 \sum_{i=1}^{i=I} Deviance\ Residual\ Lag_i + \beta_7\ Log(Offset)
\end{aligned}
\tag{1}
$$

Where *Time* indicates the daily time interval from the start of the observation period, *Intervention* is a binary term indicating the pre-intervention period (coded 0) or the post-intervention period (coded 1), *Time After Intervention* indicates the number of days after the intervention started, which takes on the value zero before the intervention, and continuous values following the intervention and *Outcome measure in Site B* represents the outcome measure in the control site.

$\beta_0$ represents the baseline level of the outcome at $t = 0$, the coefficient $\beta_1$ estimates the daily mean change in the outcome before the intervention (the baseline trend), $\beta_2$ estimates the level change in the mean outcome immediately after the intervention and $\beta_3$ estimates the change in the trend of the mean outcome after the intervention, relative to the pre-intervention trend. The sum of the estimated coefficients $\beta_1$ and $\beta_3$ represents the post-intervention slope. $Day_t$, $Week_t$, $Month_t$ variables and *Deviance Residual Lag$_i$* were also included to account for trend and seasonality, and autocorrelation respectively. The offset term log (*Offset*) represents the logged number of cages, bait stations or hanging baits, defined with respect to the outcome measure accordingly. The specific models for each outcome measure are described in S1–S3 Equations. The measure of effect for each independent linear term was the incidence rate ratio (IRR) which is the change in the daily proportion of the outcome measures with reference to the mean, associated with a unit change in the corresponding independent exposure. The IRRs were computed by exponentiating the $\beta$ value estimated from the regression analysis. An IRR value below 1.0 denotes a reduction in the outcome for each unit change in the independent variable.

Finally, backward elimination was used to achieve parsimony. All statistically significant variables from each model were simultaneously included to determine their final adjusted effect on each of the outcome measures. Statistical significance was evaluated at the 5% level. All analyses were performed using Stata 14.2 software (StataCorp, USA).

## Results

### Descriptive statistics

The study was carried out for approximately 20 weeks. There was incomplete data for 2% (3 out of 140 days) of the study duration owing to a national public holiday and 2 days of heavy precipitation events. In the treatment site (Site A), approximately 24.8kg of poisonous antico-agulant baits were employed during the intervention phase. 9 cages and 10 bait stations went missing but were immediately replaced during the next daily inspection. There was no differ-ence in the pre-intervention trends between the intervention and control sites (S1 Table).

### Outcome measure 1: Number of rodents caught

A total of 140 and 465 *Rattus norvegicus* rodents were trapped in the intervention and the con-trol sites respectively. All trapped rodents were visually identified as *R. norvegicus* (Table 1). On average, there was a 62.7% (IRR: 0.373, 95% CI: [0.224, 0.620]) reduction in the number of rodents caught in the treatment site (Site A) in the post intervention period (Fig 2 and S2 Table). There was no difference in the pre-intervention trends between the intervention and control sites (S1 Table).

### Outcome measure 2: Amount of hanging baits consumed (30g-units)

There was 25.8-unit (coefficient = -25.829, 95% CI: [-29.855, -21.804]) (equivalent to 774g) average reduction in the number of hanging baits consumed in the treatment site (Site A) fol-lowing the intervention (Fig 3 and S1 Table). There was no significant change in the trend in the control site.

### Outcome measure 3: Number of marred bait stations

There was a 61.9% (IRR: 0.381, 95% CI: [0.218, 0.665]) reduction in the number of marred bait stations in the treatment site (Site A) following the intervention (Fig 4 and S1 Table). There was no significant change in the trend in the control site.

### Post study monitoring

There was a rise in all outcome measures within a 4-month period after the conclusion of the main study phases. On average, the number of rodents caught in the treatment site increased by 13.1% (IRR: 1.131, 95% CI: [1.013, 1.263]) per day. The average number of baits consumed in the treatment site increased by 1.6-units (coefficient = 1.595, 95% CI: [0.835, 2.356]) per day. The number of marred bait stations increased by 14.5% (IRR: 1.145, 95% CI: [1.088, 1.206]) per day (Fig 5). The average rates of consumed baits and marred bait stations in this

**Table 1. Number of cages, bait stations and hanging baits deployed in the treatment site (Site A) and control site (Site B) over the study period of 20 weeks.**

| Phase | Intervention Site (Site A) | | | | Control Site (Site B) | | | |
|---|---|---|---|---|---|---|---|---|
| | No. of Cages | No. of Bait Stations | No. of 30g Hanging Baits | No. of *R. norvegicus* trapped in cages | No. of Cages | No. of Bait Stations | No. of 30g Hanging Baits | No. of *R. norvegicus* trapped in cages |
| Pre-intervention (Phase I) | 39 | 22 | 32 | 95 | 70 | 17 | 8 | 238 |
| Intervention (Phase II) | - | 37 | 60 | - | - | 48 | 65 | - |
| Post-Intervention (Phase III) | 39 | 22 | 40 | 29 | 70 | 19 | 25 | 142 |
| Post-Monitoring | 37 | 18 | 40 | 16 | 70 | 19 | 25 | 85 |

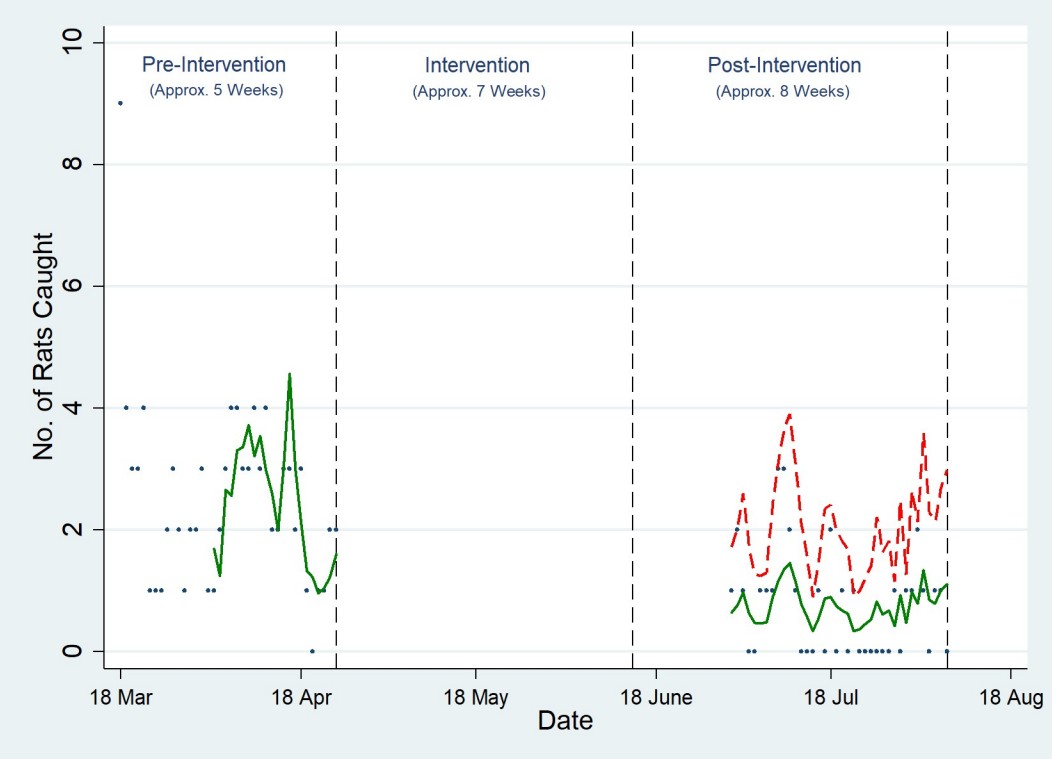

**Fig 2. Effect of intensive toxic baiting on the number of rodents caught.** The dots represent the raw data, the green line represents the fitted regression line and the red dotted line represents the counterfactual which is the predicted trend line for the number of rodents caught if there had been no intervention in place. No data was collected in the pre-baiting period during the post-intervention period.

period reached approximately 60% of that in the pre-intervention period. The rate of rodents trapped in this period did not exceed 30% of that in the pre-intervention period.

## Discussion

In this study, the effect of anticoagulant rodenticide on rodent activity was examined in order to inform strategies for reducing the transmission for rodent-borne diseases. The observed decline in rodent activity after the intervention was introduced was consistent across all three outcome measures of activity, providing strong evidence of its use as a primary intervention strategy for controlling urban rodent populations. Although other rodent activity index measures such as damage to stored products or fire incidents were not evaluated, it could be worthwhile to investigate these in the form of survey questionnaires to shop owners or local managing agents.

A before and after study conducted in New Zealand's commercial piggeries reported a decline from 27% to 5% in rodent activities attributable to the intensive anticoagulant baiting [25]. Another study conducted within almond orchards of the United States showed more than a 70% reduction in rodent activity with diphacinone grain baits [26]. Results from this study were consistent with those reported in these studies. The differences in results may be attributable to the differences in the influence of urban infrastructural factors and the availability of competing food sources [27]. Rodents are neophobic in nature and have been reported to avoid changes in location or food types [28–30]. While there was an acclimatization phase

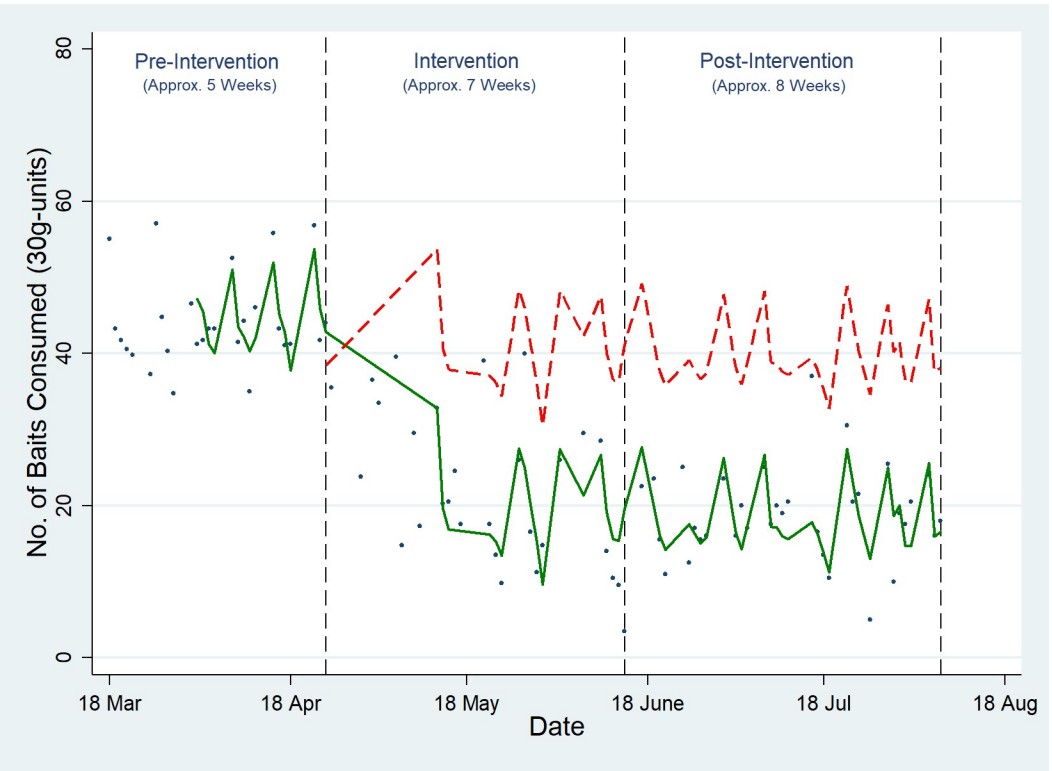

**Fig 3. Effect of intensive toxic baiting on the amount of baits consumed.** The dots represent the raw data, the green line represents the fitted regression line and the red dotted line represents the counterfactual which is the predicted trend line for the number of baits consumed if there had been no intervention in place.

for rodents to get accustomed to the cages, bait stations and hanging baits, human movement and alternative food sources in a dynamic mixed-use urban environment might have exacerbated rodent neophobia and reduced their preference for baits as their primary food source. Prior studies in Madrid and Barcelona discussed the importance of food availability and its impact on favouring rat infestations, which is correlative to housing and population density [31, 32]. In Singapore, the abundance of hawker stalls and food establishments in the study area provided strong food source competition that could possibly be more appealing than the baits used. Moreover, the studies undertaken in New Zealand and the United States utilised other forms of rodent activity indexing methods such as tracking tunnels and remote-triggered cameras, which may not be directly comparable to the measures of rodent activity levels here. It is also important to consider the type of anticoagulant used, as anticoagulant rodenticides with different active ingredients will have varying lethal doses that may affect the rate of decline of activity levels.

Physical trapping with cages or snap traps have been utilised for rodent population control. Research illustrating the sole effect of physical control on rodent activity is uncommon. A study in Hawaii reported that snap traps and self-resetting rodent traps used in combination was successful at suppressing rodent populations at <20%, which is substantially lower relative to their reference site of around 80%. Such a consequence was from 15 years' worth of rodent control. Although there were two anticoagulant bait applications, its reduction effect was minimal, which is likely due to the low rodent population sustained from the ongoing continuous trapping efforts [33]. Though trapping is effective in lowering rodent population levels, this efficacy comes from many years of continual control. Moreover, trapping requires an

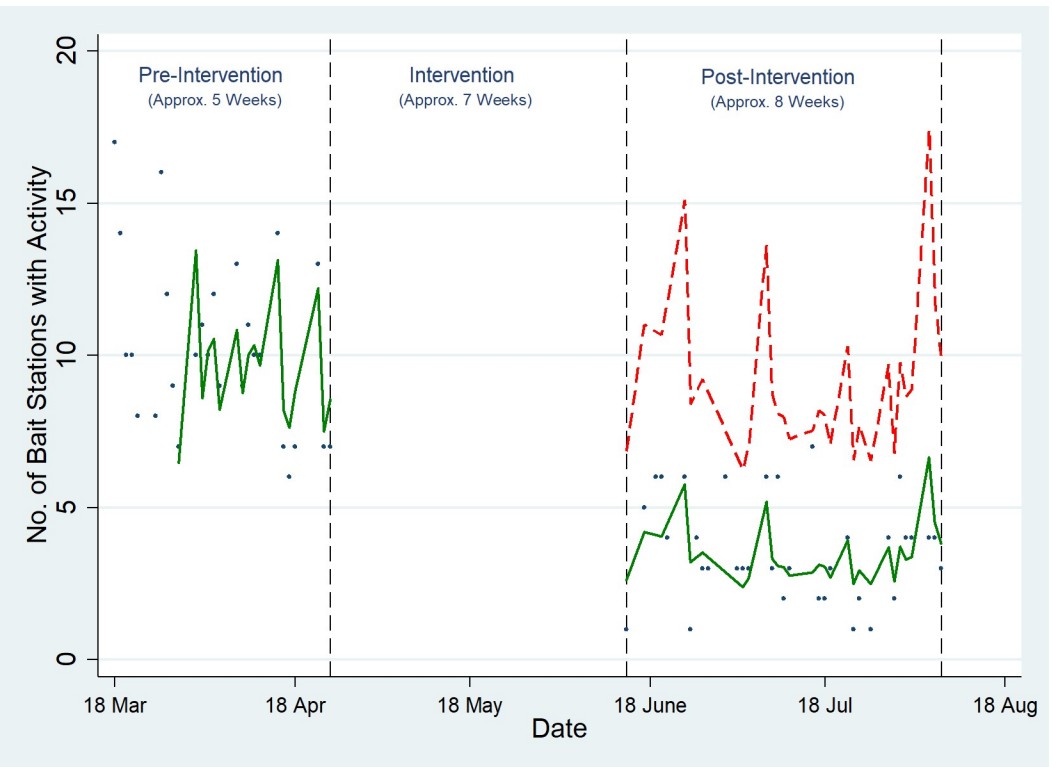

**Fig 4. Effect of intensive toxic baiting on the number of marred bait stations.** The dots represent the raw data, the green line represents the fitted regression line and the red dotted line represents the counterfactual which is the predicted trend line for the number of marred bait stations if there had been no intervention in place.

increased amount of resources in maintaining the traps and removing carcasses, which may pose as a challenge when resources are limited. If toxic baiting had been continued in this study for a prolonged duration, it is expected that the reduction in the outcome measures will begin to plateau and maintain at low levels, akin to the effects of long-term trapping.

In this study, all outcome measures demonstrated an increasing trend of rodent activity in the treatment site within four months after the intervention ceased. The most ubiquitous

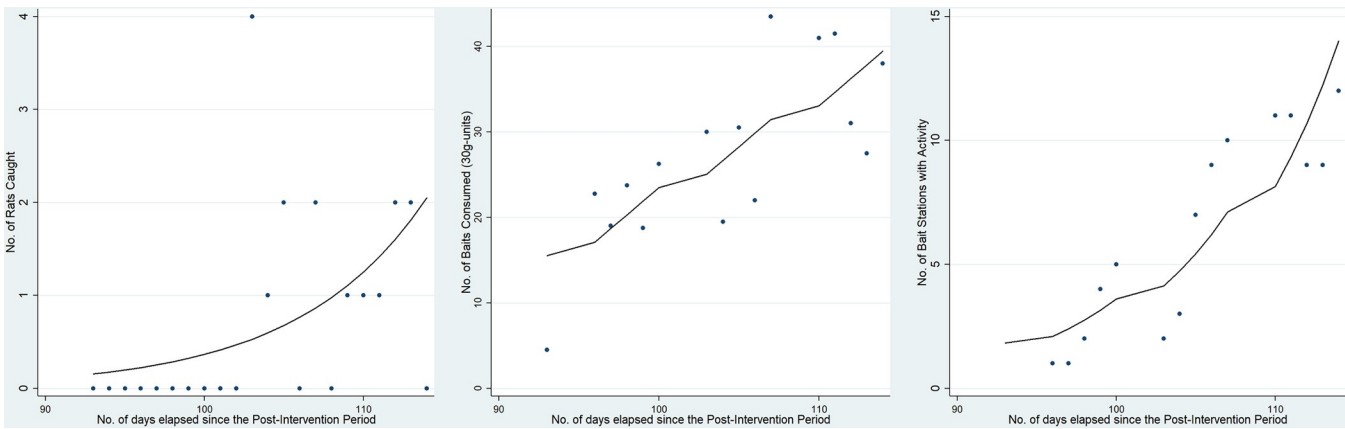

**Fig 5. Observed resurgence in rodent activity within 4 months after anticoagulant use was discontinued.** The dots represent the raw data and the black lines represent the fitted regression lines for the 3 outcome measures.

commensal rodent species in Singapore is the *R. norvegicus* [34] and due to their highly reproductive nature, a few months is sufficient for each female rodent to produce around 2 litters of pups [35]. This study did not account for the possibility of rodents emigrating from areas surrounding the site [36]. Rodent populations in neighbouring areas might have capitalised on the sudden decline of the resident rodent population to seek for increased food supply, space and new potential mates for procreation. Future studies examining rodent population resurgence and migration patterns following vector control will add to the existing literature. The observed rise in number of rodents caught in this phase exhibited a delayed increase compared to the other two outcome measures (Fig 5). A possible reason is that rodents migrating from nearby areas have yet to acclimatise to the cage traps. The rodents might have either avoided these traps completely or approached them with greater caution. The presence of hanging baits in drains and bait boxes may have been more appealing given the dark environment, allowing them to feel more at ease.

The use of an interrupted time series analysis study design was highly appropriate in evaluating and quantifying the effect of anticoagulant rodenticide control on different measures of rodent activity. The three outcome measures used provided consistent evidence of the population reductive effect of intensive anticoagulant use. The CMR approach was used to estimate the relative rodent abundance and was used as a proxy for rodent activity in this study. This approach was suitable given the difficulty in estimating the true population abundance of the rodent population. Any unmeasured secular trends that might have influenced study findings in the intervention site were accounted for with the inclusion of a control site. Though the data captured on Mondays systematically included the effects from Saturdays and Sundays, the effect of anticoagulant use on Mondays may have been underestimated since more rodents could have been trapped had the cages been reset and more baits consumed because of replenishment over the weekend. The effect estimates of the intervention are thus likely to be conservative. In addition, the data collection may have been influenced by a few instances of cages and bait stations that were tampered with or removed by members of the public. However, these were replaced immediately the next day to maximise the continuity of the observations.

The significant reductions in all outcome measures support the use of intensive rodenticide baiting in decreasing rodent populations in a dense mixed-use urban environment. Rodenticide is a common rodent control measure used globally, but it has to be used responsibly to prevent the emergence of rodenticide resistance and non-targeted secondary poisoning. This phenomenon of genetic resistance is becoming increasingly widespread due to the selection pressure from excessive anticoagulant bait use [37, 38]. Rodent control programmes may have to be adjusted through the rotational use of first- and second-generation rodenticides or pulsed baiting techniques.

This study solely employed the use of toxic baits. Although its value has been acclaimed worldwide, it should not be used as the only form of rodent control measure. A comprehensive rodent control plan aimed at reducing rodent-borne disease should be carried out in a holistic manner, with the adoption of IPM strategies. The most crucial aspect of IPM is to implement preventive measures such as proper housekeeping, storage of food and disposal of waste [39]. This reduces food availability and harbourage space, which limits the capacity of the area and restricts the maximum population density of rodents [40]. More importantly, a decline in food sources will channel rodents to feed on the deployed toxic baits, maximising its potential. Restricting rodent access and movement through the use of physical barriers and rectifying structural defects will reduce rodent activity as well. Another critical element of IPM would be health and sanitary education. This involves the distribution of educational flyers, publicising awareness posters and community engagement with residents [27]. All the above-mentioned

strategies have to be established conjointly for long-term durations in order to keep rodent populations under control.

## Conclusion

Rodent population control measures remain an essential approach in reducing the spread of rodent-borne diseases. This study demonstrated strong evidence of the effectiveness of anticoagulant use in reducing rodent activity within a mixed-used urban environment, and thus its utility as a rodent-borne disease control strategy. The expected resurgence in rodent activity after the discontinuation of anticoagulant baits reinforces the importance of continued surveillance to inform the resumption of rodent population control measures aimed at reducing rodent-borne disease transmission.

## Supporting information

**S1 Fig. Cage with bait.**
(TIF)

**S2 Fig. Hanging bait in a drain below ground.**
(TIF)

**S1 Equation. Equation for outcome measure: Number of rodents caught.**
(DOCX)

**S2 Equation. Equation for outcome measure: Amount of bait consumed (30g-units).**
(DOCX)

**S3 Equation. Equation for outcome measure: Number of marred bait stations.**
(DOCX)

**S1 Table. Pre-intervention trends of outcome measures between the intervention and control sites.**
(DOCX)

**S2 Table. Results of the effect of intensive toxic baiting on the number of rodents caught, the amount of baits consumed (30g-units) and the number of marred bait stations.**
(DOCX)

**S1 File. Raw data of outcome measures.**
(CSV)

## Acknowledgments

We express our gratitude to the Rat Control Unit of National Environment Agency for overseeing the field operations, data collection and compilation. We would also like to extend our thanks to the pest control operators who facilitated the management of the cage traps, bait stations and hanging baits on a day-to-day basis.

## Author Contributions

**Conceptualization:** Stacy Soh, Cliff Chua, Jane Griffiths, Penny Oh, John Chow, Qianyi Chan, Jason Tan, Joel Aik.

**Data curation:** John Chow, Qianyi Chan, Jason Tan.

**Formal analysis:** Stacy Soh, Joel Aik.

**Methodology:** Stacy Soh, Cliff Chua, Joel Aik.

**Project administration:** Jason Tan, Joel Aik.

**Software:** Stacy Soh, Joel Aik.

**Supervision:** Joel Aik.

**Visualization:** Stacy Soh, John Chow, Qianyi Chan.

**Writing – original draft:** Stacy Soh, Cliff Chua.

**Writing – review & editing:** Stacy Soh, Cliff Chua, Qianyi Chan, Jason Tan, Joel Aik.

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
