## [Decision Letter · Decision Letter 0]

29 Jun 2021

PONE-D-21-04665

The use of anticoagulants for rodent control in a mixed-use urban environment in Singapore: A controlled interrupted time series analysis

PLOS ONE

Dear Dr. Chua,

Thank you for submitting your manuscript to PLOS ONE. After careful consideration, we feel that it has merit but does not fully meet PLOS ONE’s publication criteria as it currently stands. Therefore, we invite you to submit a revised version of the manuscript that addresses the points raised during the review process.

Reviewer 1 has raised some significant concerns regarding your study design that need to be thoroughly addressed in your revisions. Please ensure that you provide a detailed response to each of the points raised in the reviews.

We look forward to receiving your revised manuscript.

Kind regards,

Jamie Males

Staff Editor

PLOS ONE

Journal Requirements:

Reviewers' comments:

Reviewer's Responses to Questions

**Comments to the Author**

1. Is the manuscript technically sound, and do the data support the conclusions?

Reviewer #1: Partly

Reviewer #2: Yes

2. Has the statistical analysis been performed appropriately and rigorously? 

Reviewer #1: Yes

Reviewer #2: Yes

3. Have the authors made all data underlying the findings in their manuscript fully available?

Reviewer #1: No

Reviewer #2: Yes

4. Is the manuscript presented in an intelligible fashion and written in standard English?

Reviewer #1: Yes

Reviewer #2: Yes

5. Review Comments to the Author

Reviewer #1: 1. The approach must be reversed, the vaccination plan must be implemented when control strategies fail. Environmental management should be the first option to avoid excessive growth of the rodent population, interaction with humans and the consequent risk of zoonosis due to direct or indirect circulation of pathogens.

2. Rodents, in addition to transmitting pathogens of zoonotic importance, cause significant damage and economic losses. Food contamination, broken containers and storage bags due to gnawing, as well as fires due to cable bites are some of the main consequences of the action of rodents in high densities. The use of anticoagulants is a strategy to control the population abundance of rodents, which results in a reduction in the transmission of zoonotic diseases, material damage and economic losses.

3. The experimental site and the control site are markedly different. Yellow shaded areas representing public housing apartment blocks are more abundant at the control site than at the experimental site. These differences generate noise in the results that are not contemplated. Population density data should be included at both sites.

4. Intervention: It is necessary to provide more detail on the number of bait units per bait station, to clarify what is referred to as "marred baits" in relation to rodent activity. What type of non-toxic bait was used; were they baits without an active principle?

5. Results: Table 1 is confusing. The legend of Table 1 presented in L196 is different from that of L 480, so it is not clear whether the data refer to the amount of bait consumed or the amount of baits placed. In table 1 the name of the phases does not coincide with the name that was made in the text of the ms.

6. The results show the post-intervention changes of the experimental site and the recovery of the rodent population during the monitoring phase; however, the same trend is observed in the control site without having undergone any treatment. It is important to compare the results at both sites to understand the effect of the treatment.

7. It is not understood as the effect of anticoagulant bait is not independent of rodent activity. It should clarify this sentence.

8. There are previous studies in urban environments that show the resilience of rodent populations under chemical control treatments. In all cases they agree that chemical control is only a palliative that without environmental management and health education, rodent control tends to fail. Therefore, I consider that the results obtained are not unknown by the scientific community.

Reviewer #2: Article that deals with a topic of significant interest.

6. PLOS authors have the option to publish the peer review history of their article (what does this mean?). If published, this will include your full peer review and any attached files.

Reviewer #1: No

Reviewer #2: No

---

## [Author Response · Author response to Decision Letter 0]

13 Oct 2021

Reviewer #1:

1. The approach must be reversed, the vaccination plan must be implemented when control strategies fail. Environmental management should be the first option to avoid excessive growth of the rodent population, interaction with humans and the consequent risk of zoonosis due to direct or indirect circulation of pathogens.

Our response: Yes we agree with the reviewer that environmental management should be the first option. We have deleted the following paragraph from our Introduction since our study focus related to population control rather than vaccination. We also amended out Abstract slightly.

2. Rodents, in addition to transmitting pathogens of zoonotic importance, cause significant damage and economic losses. Food contamination, broken containers and storage bags due to gnawing, as well as fires due to cable bites are some of the main consequences of the action of rodents in high densities. The use of anticoagulants is a strategy to control the population abundance of rodents, which results in a reduction in the transmission of zoonotic diseases, material damage and economic losses.

Our response: Yes we agree with the reviewer and are grateful for these important points. We have now added them into the Introduction (see text in purple).

3. The experimental site and the control site are markedly different. Yellow shaded areas representing public housing apartment blocks are more abundant at the control site than at the experimental site. These differences generate noise in the results that are not contemplated. Population density data should be included at both sites.

Our response: The size of the yellow areas are indeed different between sites. But the number of residential dwelling units are very similar between sites. In addition, the pre-intervention trends in both the control and intervention sites were similar and rate ratios of the three outcome measures between sites were non-significant (see our response to Q6 as well). This provides additional support on the similarity between sites for comparability. We recognize that readers may also raise this point. We have added the following sentence in the Methods section in lines 121-123 “There was a similar number of residential dwelling units in the intervention (550 units) and the control (517 units) sites.” We also inserted a sentence in the Results section in lines 210 to 211 “There was no difference in the pre-intervention trends between the intervention and control sites (Table S1).”

4. Intervention: It is necessary to provide more detail on the number of bait units per bait station, to clarify what is referred to as "marred baits" in relation to rodent activity. What type of non-toxic bait was used; were they baits without an active principle?

Our response: We have now provided more details on the number of bait units per bait station, and also improved the definition of “marred baits” in the Methods section. In the pre-intervention period, we used either 1 or 2 bait units per bait station. For the remainder of the study, it was sufficient to use only a single bait unit in every bait station. The non-toxic baits did not have the Bromadiolone active ingredient used in the intervention site. Please refer to the following insertions made in purple text (see attached response to reviewers document).

5. Results: Table 1 is confusing. The legend of Table 1 presented in L196 is different from that of L 480, so it is not clear whether the data refer to the amount of bait consumed or the amount of baits placed. In table 1 the name of the phases does not coincide with the name that was made in the text of the ms.

Our response: We noted the reviewer’s request for greater clarity and have now labelled the column headers in Table 1 clearly to reflect that we were referring to the number of cages, hanging baits and bait stations. We also amended the names of the respective phases to be consistent with the manuscript text. 

6. The results show the post-intervention changes of the experimental site and the recovery of the rodent population during the monitoring phase; however, the same trend is observed in the control site without having undergone any treatment. It is important to compare the results at both sites to understand the effect of the treatment.

Our response: We would like to clarify that the red dotted lines in Figures 2, 3 and 4 (see below for reference) do not represent the trend in the control group. The red dotted lines represent the theoretical absence of the rodenticide application measures in the treatment site on rodent activity, which already account for the trends in the control site. We did compare the trend for all three outcome measures in the control group and reported the results in Table S1 (see results extracted below for easy reference). Essentially, there was no significant trend in any of the three measures for the control group, therefore providing evidence that the reduction in the measures at the intervention was related to the anticoagulant treatment rather than any changes in the secular trend (as represented by the control site). To improve clarity, we have now added “There was no significant change in the trend in the control site.” at the end of the results for each of the three outcome measures. In addition, we computed the rate ratios (RR) and corresponding CIs for the normalised outcome measures in the pre-intervention period. The normalised rate is the average rate of the outcome measure over the pre-intervention period. The RR is defined as: (see attached document). The 95% confidence intervals for the RRs overlapped with 1 and were non-significant for all three outcome measures. This supports that both the treatment and control site were similar prior to the intervention.

7. It is not understood as the effect of anticoagulant bait is not independent of rodent activity. It should clarify this sentence.

Our response: We agree with the reviewer that the effect of anticoagulant bait is not independent of rodent activity. The reviewer may have been referring to the first sentence of the Discussion section. We amended that sentence as follows to improve clarity: “In this study, we examined the independent effect of anticoagulant rodenticide on rodent activity.”

8. There are previous studies in urban environments that show the resilience of rodent populations under chemical control treatments. In all cases they agree that chemical control is only a palliative that without environmental management and health education, rodent control tends to fail. Therefore, I consider that the results obtained are not unknown by the scientific community.

Our response: We agree with the reviewer that previous studies have quantified the effects of chemical control on rodent populations. Studies in South-East Asia are more limited though and we were able to quantify the impact, which can be used to inform the allocation and augmentation of resource plans to improve rodent control. We have made the following amendments to provide a more precise justification for our study:

In the Abstract, we amended the Introduction to “Studies assessing quantifying the impact of anticoagulant bait use on rodent populations are scarce in tropical settings.”.

In the Introduction, we amended the last sentence as follows: “In this study, we assessed quantified the impact of intensive anticoagulant use as an urban rodent population control measure, in order to inform the implementation of rodent population control strategies in Singapore.”

---

## [Decision Letter · Decision Letter 1]

28 Feb 2022

PONE-D-21-04665R1The use of anticoagulants for rodent control in a mixed-use urban environment in Singapore: A controlled interrupted time series analysisPLOS ONE

Dear Dr. Chua,

Thank you for submitting your manuscript to PLOS ONE. After careful consideration, we feel that it has merit but does not fully meet PLOS ONE’s publication criteria as it currently stands. Therefore, we invite you to submit a revised version of the manuscript that addresses the points raised during the review process.

We look forward to receiving your revised manuscript.

Kind regards,

Yasmina Abd‐Elhakim

Academic Editor

PLOS ONE

**Reviewers' comments:**

**Reviewer #1: **The explanations and modifications made to the paper by the authors were accepted without inconvenience but pending issues that require attention remain.

IRR defined as Incidence Rate Ratio is only described in the annex available only to reviewers. Please include your calculation in the body of the paper (statistical analysis).

Line 246-247: Since the circulation of pathogens was not specifically evaluated before and after the intervention, it is recommended to also include other problems that can generate such as damage to stored products, fires, etc.

line 255. It is recommended to review the paper of Fernandez et al. 2007 that provides details in this regard.

Line 265: It is also important to consider in order to compare the responses to the use of rodenticides which is the active principle since they present differences in their lethal doses.

Line 314: Health education should be considered as a fundamental component of IPM, since without health campaigns it has been shown that all the control strategies that are implemented tend to fail (read the attached bibliography).

**Reviewer #3: **Despite the importance of the topic of this manuscript, but the authors need to fix several points as follows:

1. The information present in the manuscript are overlapped and need to be rearranged.

- Firstly, the introduction is very short and not properly cover the topic but most required information need are present in the material and methods section. Hence, the authors should transfer the background on the food establishments (lines 74-86) to the introduction. Also, the background on the rodenticide used (lines 114-118) should exist in the introduction.

- All possible interpretations of the study findings should exist in the discussion section. E.g. the justification of using the capture-mark-recapture (lines 130-133), the justification of using an interrupted time series analysis (lines 150-153), the justification of inclusion of a control group (lines 154-157), …and so on.

2. The manuscript needs proofreading. Writing style should be formal from the third-person perspective. Do not use we, they, or ours. Please check the whole manuscript.

3. The legend of Table 1, the location of the study and the duration should be mentioned.

---

## [Author Response · Author response to Decision Letter 1]

1 Apr 2022

The response to both reviewers can be found appended inside the "Response to Reviewers" word document.

---

## [Decision Letter · Decision Letter 2]

18 Apr 2022

The use of anticoagulants for rodent control in a mixed-use urban environment in Singapore: A controlled interrupted time series analysis

PONE-D-21-04665R2

Dear Dr. Chua,

We’re pleased to inform you that your manuscript has been judged scientifically suitable for publication and will be formally accepted for publication once it meets all outstanding technical requirements.

Kind regards,

Yasmina Abd‐Elhakim

Academic Editor

PLOS ONE
---

## [Editor Report · Acceptance letter]

12 May 2022

PONE-D-21-04665R2 

The use of anticoagulants for rodent control in a mixed-use urban environment in Singapore: A controlled interrupted time series analysis 

Dear Dr. Chua:

I'm pleased to inform you that your manuscript has been deemed suitable for publication in PLOS ONE. Congratulations! Your manuscript is now with our production department. 

Kind regards, 

on behalf of

Dr. Yasmina Abd‐Elhakim 

Academic Editor

PLOS ONE